# The Roles of Microcredit in Informal Housing in the Future—A Case Study in Hong Kong

Chung-Yim Yiu [1],* and Ka-Man Leung [2]

1    Department of Property, The University of Auckland Business School, Auckland 1142, New Zealand
2    Department of Real Estate and Construction, The University of Hong Kong, Hong Kong SAR 999077, China
*    Correspondence: edward.yiu@auckland.ac.nz

**Abstract:** Microcredit is usually used to support employment, poverty reduction, women empowerment, etc. It is rare to have studies on using microcredit to help residents in informal housing to improve their residential mobility. This study is a novel attempt to explore the roles of microcredit in informal housing in the future by taking Hong Kong as a case study. This study aims to investigate whether microcredit affects the relocation decisions of low-income tenants in informal housing markets by using a quasi-experimental approach. A microcredit scheme for this purpose was first proposed, pitched for funding, and then implemented by a non-governmental organization (NGO) in Hong Kong to provide interest-free loans for households living in sub-divided units (SDUs) to pay for rental deposits. Interviews were conducted with SDU households. The results show that the microcredit scheme is conducive to the relocation decisions of low-income households, especially in emergency cases. This study shows the key role of microcredit in empowering low-income households in their relocation decisions, and it can make a difference to the future informal housing markets in the world.

**Keywords:** residential mobility; financial constraints; microcredit; informal rental housing markets; sub-divided units; Hong Kong

## 1. Introduction

The COVID-19 pandemic has exposed the vulnerabilities of residents living in informal housing, including the health risks due to the overcrowding conditions, a higher risk of eviction and rental increase [1]. The long-term solution may be an increase in affordable housing and social housing, but a short- to medium-term solution is to empower residents of informal housing to bargain with landlords. Unfortunately, most residents of informal housing earn very low income, and they have very few choices in the rental housing markets. Some of them have to bear unreasonably high unit rents and surcharges of utility bills due to their lack of resources to move their homes. It can result in spatial mismatch or spatial lock-in. How to make cities and human settlements inclusive, safe, resilient and sustainable is the goal of the United Nations' Sustainable Development Goal 11 (SDG 11). This paper therefore aims to study the role of microcredit in helping informal housing residents' residential mobility.

Microcredit is considered one of the major poverty reduction strategies [2]. It has been intensively studied and found to have a great role in reducing poverty [3], especially in emergencies or natural disasters [4]. However, most of the studies are in developing countries on employment generation, poverty reduction, and women empowerment. It has rarely been examined in developed cities until the recent decade. For example, Biggers et al. [5] studied the impacts of the Community Empowerment Fund on accommodating the homeless to permanent housing in the U.S. Yet, they are mostly focused on homeownership and running businesses. This study, however, is a novel attempt to launch and investigate the effects of a microcredit scheme on the residential mobility of informal housing tenants in Hong Kong.

Residential mobility has found to be essential for households to minimize spatial mismatch [6], and for the society at large to achieve an efficient labor market [7]. Residential immobility is commonly claimed to be caused by financial constraints. For example, homeowners can be spatially 'locked-in' due to financial constraints associated with housing market declines such as falling home prices or rising interest rates [8–10]. However, most studies on residential mobility focused on homeowners, probably because renters are usually more mobile. Causa and Pichelmann [11] showed that, in OECD countries, renters in private housing markets were almost 5.6 times more mobile than homeowners. Almost 80% of the renters in private housing markets moved their homes in the past 5 years in the U.K., the U.S. and Australia. However, this figure concerns tenants in formal housing markets, the phenomenon of spatial lock-in of low-income renters, especially households in informal housing markets, is largely neglected.

More importantly, the causality between financial constraints and residential mobility has not yet been established. Causa and Pichelmann [11] contended that 'causality here is not established because of endogeneity and self-selection biases' (p. 39). Lui and Suen (2011) [12] also shared the same caution by highlighting that 'in the absence of experimental data or convincing instrumental variables, a causal interpretation of the [residential mobility] results should be adopted with some caution' (p. 21). The caution of endogeneity is that housing choice is generally associated with individual characteristics that may generate lower mobility. In other words, the observed association between financial constraints and residential immobility can be caused by a confounding factor. This study, therefore, aims to apply a quasi-experimental approach of microcredit to study the impact of a microcredit scheme on residential mobility in the informal rental housing market in Hong Kong. It applies an intervention approach to study how the change in financial constraints affects the decisions of relocation of low-income households living in informal housing.

A field experiment with randomized control trials is one of the best ways to test causality, but the difficulty of doing randomized trials in economics is well recognized. That is why sometimes only quasi-experiments can be conducted. The introduction of the field-experimental approach by Michael Kremer, Esther Duflo, and Abhijit Banerjee, the Nobel Laureates in 2019, since the 1990s has transformed the research methods in economics. In the past, economic studies are mostly based on datasets of past events or laboratory-based experiments. One of the merits of using a field experiment is the market-based intervention (treatment) that can be introduced to determine whether the treatment is a real cause or not. This study attempts to use a market-based intervention approach to investigate the impact of financial constraints on residential mobility. The research question is whether the residential mobility of low-income tenants in informal housing markets is affected by their financial constraints in paying rental bonds, agent commission, etc. The intervention introduced is a novel microcredit scheme—a relocation assistance scheme, which is a pilot scheme firstly introduced in Hong Kong by an NGO for helping low-income tenants to afford the expenses of residential relocation, which include rental deposit, agent commission, and relocation transportation costs.

It is a common practice in the world that tenants are required to pay one to two months' rents as a rental deposit, it will either be kept by landlords or a third-party institute until the end of the tenancy. It may not be a concern for rich or middle-income families, but it can render low-income and no-saving families to be spatially locked-in and they may have to accept unreasonable terms of renewal offered by the landlords. Such a situation of spatial lock-in of tenants is more commonly found in informal housing markets as there are more low-income households. A typical type of informal housing in Hong Kong is sub-divided unit (SDU), which will be described in the ensuing section.

The arrangement of this paper is as follows. Section 2 provides the background information of SDU, rental bond system, and the microcredit scheme for low-income families in Hong Kong as well as a literature review. Sections 3 and 4 discuss the research methods and the results. Section 5 concludes.

## 2. Literature Review

### 2.1. Sub-Divided Units (SDUs) in Hong Kong

Hong Kong has been regarded as a severely unaffordable housing market [13]. Even though there is a public rental housing (PRH) scheme, the Hong Kong Housing Authority (2022) [14] reported that there were about 144,200 PRH general applications, resulting in an average waiting time for allocation to be 6.0 years. The shortage of housing units in formal housing sector creates overwhelming demand for alternative housing, especially sub-divided units (SDUs), which involve subdivision of a medium-sized residential quarter into two or more rooms for rental purpose [15]. According to the latest government's estimates, there were about 127,500 inadequately housed households demanding for housing conditions improvement. Among them, about 75% of them lived in SDUs [16]. SDUs have gradually become a key category of informal housing in Hong Kong. Since SDUs are extremely small in size but very expensive in unit rent per square meter, much attention has been drawn to the housing affordability and living conditions of the residents. Very few studies focus on the causes of residential immobility of SDUs' tenants.

According to the 2016 Population By-census, 27,112 quarters were subdivided into SDUs in Hong Kong [15]. On average, each quarter was subdivided into 3.4 SDUs. The median floor area of SDUs was 107.6 square feet (~10 square meters) which was only 25% of that of all domestic households in Hong Kong. The median per capita floor area of SDUs was 62.4 square feet (~6 square meters) comparing to 161.5 square feet (~16 square meters) in the formal housing markets. SDUs provided shelters for 91,787 households and 209,740 persons, which were about 2.9% of the population of the whole territory.

As an illustration of the layout of SDUs, Figure 1 shows a typical apartment flat and its mirror image flat being subdivided into four SDUs with floor areas of only about 10 sq. m. each. This shows the most typical layout of SDUs in Hong Kong, in which a Chinese Tenement (Whole Flat) with two staircases was subdivided into several SDUs. The figure shows one whole flat without subdivision (the unit on the right-hand side) and a mirror image of the flat being subdivided into four SDUs, viz. SDU1, SDU2, SDU3, and SDU4 (the unit on the left-hand side). It shows that the living room is subdivided into three SDUs, and the original kitchen and toilet are converted into another SDU. In this typical case, all the SDUs have independent toilets and windows, as shown, but in some cases, the subdivisions may not be able to provide an independent toilet and/or a window for each SDU. The associated building works in the subdivision can result in various irregularities, posing building safety problems.

The preliminary survey results in 2019 suggested that the residential mobility of SDU tenants is lower than their counterparts in private formal housing. Policy 21 Limited [17] found that 47.4% of SDU tenants had moved in the past three years, compared with 60.2% of tenants who had moved their homes in the past five years in the private formal housing markets of Hong Kong [12]. The moving costs of tenants did not receive much attention in the previous studies. Tenants are assumed to have financial ability to afford the expenses involved. However, the moving costs could be a burden to low-income households living in SDUs. According to the latest census [15], the median monthly household income of SDU tenants was HKD 13,500 (USD 1730) in 2016, which was only 54.0% of the median income of all domestic households in Hong Kong. Moreover, a survey conducted by Caritas Development Project for Grassroots of Organizations [18], in which 70.5% of the respondents were SDU households, found that 76.0% of the respondents were waiting for public rental housing allocation and over 50.0% of the respondents met the requirements for Low-income Working Family Allowance Scheme.

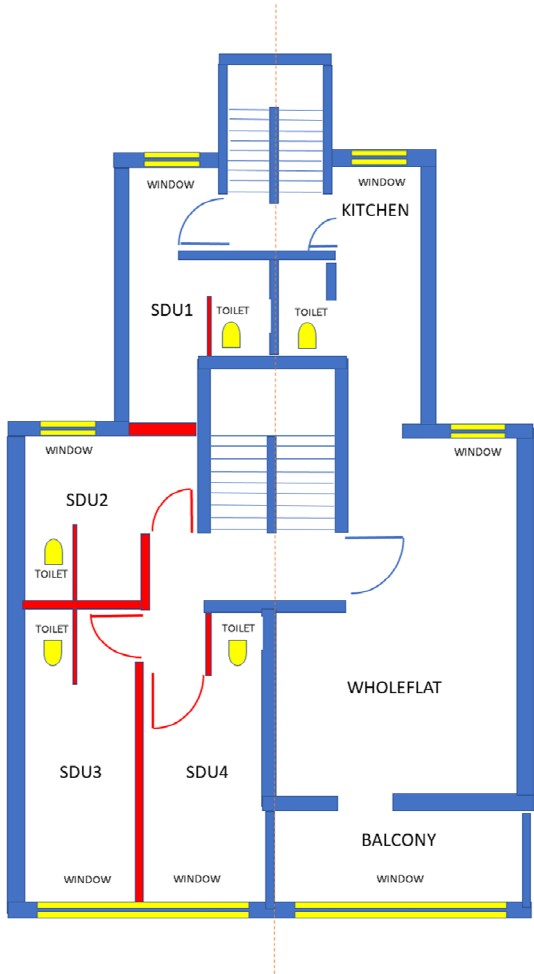

**Figure 1.** A floor plan of a typical flat and its mirror image flat being subdivided into four SDUs. Notes: The figure shows one whole flat without subdivision (the unit on the right-hand side) and a mirror image of the flat being subdivided into four SDUs, viz. SDU1, SDU2, SDU3, and SDU4 (the unit on the left-hand side). It shows that the living room is subdivided into three SDUs, and the original kitchen and toilet are converted into another SDU. In this typical case, all the SDUs have independent toilets and windows as shown, but in some cases the subdivisions may not be able to provide an independent toilet and/or a window for each SDU. The associated building works in the subdivision can result in various irregularities posing building safety problems. Source: drawn by the authors.

The building conditions of many SDUs are below regulatory standards. Structurally, SDUs are formed by subdividing an apartment unit into two or more independent units for rental purposes. It is common that non-structural partition walls in the original units are removed while new ones are erected to form the SDUs. Internal drains may be added or altered for installing independent toilets and/or kitchens (according to the information provided by the Buildings Department [19], the common irregularities of SDUs in comparison with the regulatory standards of formal housing are "removal of the original non-structural partition walls, erection of new non-structural partition walls, installation of new toilets and kitchens, alteration or addition of internal drains, thickening of floor screeding to accommodate the new/diverted drain pipes, addition of unauthorized door openings which contravene the regulations on means of escape, additions of openings for ventilation affecting the integrity of fire resisting construction, improper drainage works causing water seepage, and overloading of the building due to addition of non-structural partition walls and thickened floor screeding"). These alterations are mostly carried out

without undergoing formal regulatory procedures (it is reported that only 2% of the new SDU properties submitted Certificate of Completion after subdivision works in 2016, implying that 98% of the newly converted SDU properties may involve irregularities [20,21] (Buildings Department, 2019; Liber Research Community, 2018)). Together with a lack of maintenance, they often result in water seepage, concrete spalling, fire hazards, and poor ventilation.

### 2.2. Rental Bond System in Hong Kong

It is globally common that landlords require tenants to pay a rental bond (or a rent deposit) for soliciting tenure security of a housing tenancy and as a protection of the landlord's rights. Similarly, tenancy agreements in Hong Kong include a payment of rental deposit in the amount equivalent to one or two months' rent as an "earnest of performance" of obligations under the tenancy agreement (Community Legal Information Centre, 2021) [22]. The property owners usually keep the rental deposit and return it to the tenants when the tenancy contract expires, after deducting outstanding liabilities. In other words, a tenant has to bear the deposits of two tenancies before moving from one housing unit to another in the transition period. The tenant also has to bear relocation costs and agent commission fee, if applicable, as well.

Paying rental deposits may not be a concern for high- or middle-income tenants, as they have either enough savings or access to personal loans to pay the deposits. However, it is not the case for low-income or no-income families. Bank loans are usually not accessible to low-income people due to the risk assessment criteria on repayment capacities. Under this circumstance, the financial constraints may limit households' relocation decisions even when they face unreasonable rental increases or building conditions issues. They may not be able to move to another residence that charges a more reasonable rent and/or provides a better living environment if they cannot afford to pay the rental deposit. The only choice is to stay in the same unit and renew the existing tenancy contracts—i.e., residential immobile or spatial lock-in.

If the argument is correct, an interest-free short-term loan designated for paying the rental deposit and relocation expenses (such as a microcredit scheme) should enable the needy tenants to move more freely. In contrast, if the immobility is not caused by financial constraints but by other confounding factors of household characteristics, a microcredit scheme would not make any difference to their relocation decisions.

### 2.3. Charity Support to Low-Income Families in Hong Kong

In Hong Kong, there are three relevant schemes supporting low-income households in SDUs but with various limitations. The first one is the Relocation Allowance for Residents of Illegal Domestic Premises in Industrial Buildings scheme. It is only applicable for tenants (1) living in industrial buildings and (2) are forced to move out as a result of the Buildings Department's Enforcement Action. Thus, its target is very limited and could not help SDU tenants living in residential (non-industrial) buildings in general. The second scheme is the Assistance Programme to Improve the Living Environment of Low-income Sub-divided Unit scheme, which is funded by the Community Care Fund. Yet, its scope is confined to providing a one-off subsidy in kind for (1) carrying out minor improvement/repair works, (2) purchasing furniture and household goods, and/or (3) pest control services. In other words, it does not support home relocations or rental deposit uses. The third scheme is the Rainbow Fund, which helps individuals and families on the verge of a financial crisis [23]. Each successful applicant will receive a grant ranging from HKD 2000 (USD 256) to HKD 10,000 (USD 1282), but the grant is usually not enough for paying two-month rental deposit, as the purpose of the grant is not for household relocation. Moreover, households that receive other public funds, such as the Comprehensive Social Security Assistance (CSSA) Scheme, are not eligible for the Rainbow Fund. Since many low-income households living in SDUs receive public assistance, they are usually not eligible to apply for the Rainbow Fund. For example, seven out of the nine respondents are recipients of

CSSA, social security for the financially vulnerable. It should also be noted that charity grants are often in smaller amounts, and they are usually provided by the government as social welfare. A microcredit scheme, on the other hand, is a loan requiring repayment with or without interest payments.

### 2.4. Residential Mobility and Financial Constraints

When home occupiers are spatially locked-in (i.e., residential immobile) due to certain constraints, it accumulates spatial mismatches [24], resulting in a less efficient labor market and does harm to the economic growth of the city at large [7]. Kain [25] raised the issue of "spatial mismatch" for people living far away from their jobs. However, the reasons why people do not move when there are spatial mismatches can vary. The most common explanation is the financial constraints of the residents [8–10] or retaining subsidies in social housing [26]. However, there can be many other reasons, such as personal or family preferences, school zones, or social assets consideration. Both Causa and Pichelmann (2020) and Lui and Suen (2011) [11,12] raised the concern of endogeneity and self-selection biases in explaining residential mobility by financial constraints, especially when residential mobility is measured by the length of residence or the likelihood of a household's relocations [27,28]. In other words, previous studies relying on a simple econometric approach could not establish causality between financial constraints and residential mobility, as there has been no experimental study using an interventional approach on the causality test.

Moreover, most of the previous studies on residential mobility focused on homeowners probably because their mobility is, in general, lower than that of renters. However, if the financial constraint is a plausible cause of immobility, it is reasonable to expect lower mobility among low-income renters. Sánchez and Andrews [28] is one of the exceptions that studied the mobility of renters, and they found that "reducing rent controls and changing the bargaining power balance between tenants and landlords could increase residential mobility" in OECD countries. However, they found no effect on real estate and registration fees. They used a cross-country probit regression analysis, which included general households. Their incomes might be high enough to afford the real estate fees such that the fees do not deter them from moving, but it may not be the case for low-income households living in informal housing.

### 2.5. Microcredit

Many previous studies on microcredit are on the effectiveness of microfinancing in empowering and enhancing social mobility [29–32] rather than residential mobility. Microcredit programs are largely reported to assist numerous poor people and relieve their financial difficulties in developing countries. For instance, the National Bank for Agricultural and Rural Development (2020) [33] of India pioneered the bank linkage model, which positions the Self-Help Groups as financial intermediaries to enable the flow of bank loans to poor farmers without physical collateral. However, some studies raised concerns about misuse [34], over-borrowing [35], and whether it should be interest-bearing or interest-free [36–38]. Roxas and Ungson (2012, p. 11) [39] raised the question of "whether financial success or non-financial objectives (i.e., helping the impoverished) should be the primary measure of new microfinancing ventures". This study, therefore, focuses on a designated interest-free loan for paying rental deposit and related tenancy expenses only and measures the success of the microcredit scheme by the actual changes in the residential mobility of the tenants.

Housing microcredit schemes were not common in developed cities until the recent decade, and there has been no study on using microcredit schemes as an intervention approach to study the impact of financial constraints on residential mobility. Most of the studies on housing microcredit are about "the provision, improvement or adaptation of housing" [40]. Biggers et al. [5] examined an innovative new program's ability to assist those experiencing homelessness to attain and retain permanent housing. They raised a vicious cycle theory between housing, employment, and assets that "people cannot obtain

housing without assets and cannot obtain assets without employment. However, they often find that they cannot obtain employment without housing". It is because it requires some "start-up costs" for moving into a housing unit, including security deposits, the first month's rent, furniture, moving costs, etc. It provides the theoretical framework for the association between financial constraints and residential mobility.

This paper studies the first pilot microcredit scheme in Hong Kong designated for home moving. As a pilot scheme for testing the financial constraint factor, the scheme is set to be interest-free, and no administrative fee is charged to the borrowers. These conditions help simplify the procedures and exclude other financial reasons for tenants' immobility.

## 3. Materials and Methods

This study adopts an intervention approach by introducing a novel microcredit scheme for relocation expenses, including rental bond payments, to low-income households in Hong Kong. It provides a novel case study to examine the impact of financial constraints on the residential mobility of households in SDUs, thus analyzing whether a microcredit scheme would affect tenants' relocation decisions.

The preparation stage started in 2017. A roundtable meeting with NGOs to discuss SDU issues, including microcredit schemes, was held. Then, one of the NGOs, HKSKH Lady MacLehose Centre Group & Community Work Unit (2018) [41], managed to raise a seed fund from a private sector donation to launch the program called "Relocation Assistance Scheme", which provides a wide range of assistance to SDU households, including a pilot microcredit-like scheme to assist low-income tenants to pay relocation expenses and rental deposit as well as other related tenancy expenses. As a pilot scheme, the funding is sufficient to provide, in each batch, financial assistance to about ten SDU households in need when they lack money to move to another residence. The households have to repay without interest in installments. This is the first pilot attempt at housing microcredit in Hong Kong organized by an NGO, the loan amount for each household is about HKD 8000 (USD 1026), and the loan repayment period is usually split into eight months. Since the purpose of the scheme is to facilitate household relocation, the loan amount granted can be up to a two-month rental deposit. All the available funds of about HKD 126,000 (USD 16,200) were lent out to 16 households within months. Most of them repaid their loans, while a few cases received repayment period extensions or defaulted.

The interim reviewing stage started in 2021. As the microcredit scheme was launched in 2020, the interviewees are the nine inquirers of the microcredit scheme in the first batch. They are invited to be interviewed by means of a semi-structured survey to understand the reasons for relocation and the impacts of the microcredit. However, this initial study uses a qualitative approach, and the sample size is small.

To evaluate the effects of the microcredit scheme on the relocation decisions of households living in SDUs, questions are divided into six parts. Questions in Part One concern households' experience of using microcredit schemes and how they find a way to afford the moving costs if the loan is not available. Parts Two and Three are the focuses of this paper, which are about how microcredit affects their relocation decisions and how the microcredit scheme affects their living environments and rents.

Since SDU household data are not publicly available, they are collected through online or face-to-face interviews. Most of the interviewees lived in the Kwai Tsing and Tsuen Wan districts, where 4.1% and 6.9% of residential flats were found to be subdivided to accommodate 10,222 households [15]. In this study, tenants who could not relocate without financial support are defined as spatially lock-in tenants. Their residential immobility can result in a spatial mismatch. The spatial mismatches of households in these two districts can be partially reflected by the high rates of cross-district commuting of the working populations (63.7% and 60.3%), with the higher one ranking the fourth highest in Hong Kong [42].

The authors of this study prepared the questions. The NGO, which operates the microcredit scheme, was responsible for inviting the interviewees and arranging the venues

for the interview. Each interview was about 45 to 60 min long. We interviewed nine SDU tenants, with six of them applying and receiving microcredit and three of them deciding not to apply. All of them had moved homes before the interviews. All the respondents admitted that one of the major considerations of moving homes is financial constraint. The reasons the three respondents did not borrow the loans are that they applied for the Rainbow Fund or borrowed money from relatives. In other words, they were also facing financial constraints in paying rental deposits but preferred other repayment terms.

Table 1 presents descriptive statistics of the respondents' housing and household characteristics. If the cost of moving requires paying two months' rent for the security deposit, one month's rental pre-payment, agent's commission fee (usually half month's rent), etc., the total cost of relocation may amount to about twice the household's monthly income. For the borrowers, the average monthly SDU rent of the current tenancy is HKD 4660 (USD 597), while that of the previous tenancy was HKD 4300 (USD 551). The average SDU floor area drops from 158.7 square feet to 96.0 square feet. The average household income is only HKD 7875 (USD 1010), and the rent-to-income ratio is about 60% which is almost double that of the overall average of SDU households. The summary statistics also indicate that the purpose of moving is not for bigger housing units or reducing rents.

**Table 1.** Respondents' household and housing characteristics before and after relocation.

| Variable | Before Relocation | | After Relocation | |
|---|---|---|---|---|
| | Borrower (*n* = 6) | Non-Borrower (*n* = 3) | Borrower (*n* = 5) | Non-Borrower (*n* = 2) |
| Average Monthly Household Income, Including CSSA * (HKD) | 7875 | 8933 | 6250 | 8900 |
| Average Household Size (No. of Persons) | 2.3 | 2.0 | 2.2 | 2.0 |
| Average SDU Housing Size (Floor Area in Square Feet) | 158.7 | 182.5 (*n* = 2) | 96.0 (*n* = 5) | 72.5 (*n* = 2) |
| Average Housing Rent per Month (HKD) ** | 4300 | 4600 (*n* = 2) | 4660 (*n* = 5) | 3750 (*n* = 2) |

* Among all the respondents, 77.8% receive Comprehensive Social Security Assistance (CSSA), which is a public safety net for poor households. ** Before relocation, one non-borrower household lived in a whole flat. After relocation, one borrower household moved to community housing, and another non-borrower moved to public rental housing.

*Semi-Structured Interviews*

Since this is a novel pilot trial of using microcredit to support household relocations and the study is also the first one in the field, semi-structured interviews with the individual applicant (both successful and failed cases) are considered as the most appropriate research method to explore the research question. It allows better engagement between the participants, including tenants, social workers of the NGO, and researchers, to build confidentiality and trust. Interactions with open-ended questions enable the interviews to yield findings in sufficient depth and breadth. The following shows some of the open-ended guided questions:

1. Tell us about your relocation experiences with the microcredit scheme.
2. How does the microcredit scheme affect your decision of tenancy renewal/relocation?
3. If there is no such scheme, what would be your choice, and how will it affect your family?
4. What are the differences in the living conditions and the monthly rents, comparing the new tenancy and the previous tenancy?
5. Does the microcredit scheme enable you to have better bargaining power with the landlord?

In the interviews, we also asked some 5-point Likert scale questions to rank the major factors determining their decisions to relocate as follows (Table 2):

**Table 2.** The 5-point Likert scale questions for ranking the major factors determining relocation decisions.

| How Important Are the Following Factors to Determine Your Decisions of Home Relocation? | Please Circle (1 = Not Important; 3 = Important; 5 = Very Important) | | | | |
|---|---|---|---|---|---|
| 1. The conditions of the present SDU | 1 | 2 | 3 | 4 | 5 |
| 2. Neighbors | 1 | 2 | 3 | 4 | 5 |
| 3. The location of the present SDU | 1 | 2 | 3 | 4 | 5 |
| 4. Rent | 1 | 2 | 3 | 4 | 5 |
| 5. Electricity and water bills | 1 | 2 | 3 | 4 | 5 |
| 6. Tenancy matters | 1 | 2 | 3 | 4 | 5 |
| 7. Financial availability for paying the rental deposit and the prepaid one month's rent | 1 | 2 | 3 | 4 | 5 |
| 8. Cost of relocation | 1 | 2 | 3 | 4 | 5 |
| 9. Change of family conditions | 1 | 2 | 3 | 4 | 5 |

## 4. Discussion

### 4.1. Reasons for Moving Homes

Traditional models of residential location choice study formal housing markets only. Their findings usually emphasize location, accessibility, and rent. However, our study found that SDU tenants not only care about housing rent but also interior conditions and the cost of relocation. By means of detailed interviews, we identified that interior conditions do not merely mean housing size or renovation, but they also refer to health and safety issues, such as water seepage or leakage and concrete spalling. These health risks are particularly intense during the COVID-19 pandemic.

The top five most important reasons for relocation from the respondents' responses are unaffordable rents, interior conditions, cost of relocation, utility bills, and location. The average scores for the five factors are 4.8, 4.7, 4.2, 4.1, and 3.6 out of 5.0, respectively. Among these five most important factors, three of them are about costs, including rents, cost of relocation, and utility bills. It reflects that housing cost is the major concern of low-income families.

More importantly, one of the most important reasons for moving homes is interior conditions. The building conditions of some SDUs that were previously occupied by the respondents are very poor, if not uninhabitable. For example, one respondent said that water seepage was very serious in the previous SDU that they occupied. There was an incident that some of the electrical appliances were damaged by water seepage, causing electricity leakage. The decision to move home, in this case, is an emergency relocation. There were also two emergency cases in which the tenants had to leave involuntarily and urgently. One respondent said he received a demolition order from the government and was coerced to leave immediately. Another case involved an incident of a water pipe bursting, which made the SDU not habitable. These tenants could become homeless if there is no financial support to pay for the deposit of renting another housing unit. In contrast, some respondents indicated that they would have to either stay in the previous SDUs longer to save enough money or to ask relatives for help if there is no other financial support. These cases reflect that the residential immobility situation raised in the literature can be underestimated as many households choose to stay in the SDUs due to the financial difficulties in paying relocation costs unless emergency relocations or involuntary relocations are required.

It is also noted that the frequency of relocation of SDU households is relatively high. In our samples, the average period of staying in the previous SDUs was only 15.2 months, indicating that most of them moved out without completing the 2-year tenancy period. It

probably reflects the information asymmetry on building conditions between landlords and tenants. For example, water seepage can hardly be identified by visual methods if it is covered up and is not surveyed during rainy days. Furthermore, due to a very limited supply of SDUs, landlords could ask for a substantial rental increase upon tenancy renewals. Tenants in SDUs are therefore expected to have a higher frequency of moving. However, our results found that it is the cost of moving that deters them from moving. The keen applications for the microcredit-like Relocation Assistance Fund and the Rainbow Fund show a strong case that microcredit can help residents of informal housing by enhancing residential mobility. The results are more pronounced in emergency or involuntary relocations.

### 4.2. Effects of the Microcredit Scheme on Borrowers' Rental Decisions

The microcredit-like relocation assistance scheme is found to be particularly effective and important in supporting emergency or involuntary relocations of low-income households. For example, in the two cases involving removal orders from the government, the respondents had to leave the residence in two weeks' time. Within such a short period of time, it was hard for them to gather enough money to pay for all the relocation expenses. Another three cases are involved with water leakage problems and overcrowding issues. (There is one case that the interviewee moved to a community housing unit. The monthly rent equals the maximum level of the rent allowance under the CSSA scheme or not exceed 25% of household income (Community Housing Movement, 2021)) [43]. Inferior building conditions is one of the major urges of their moves. Here we quote the responses of three interviewees on what they would do if there were no microcredit scheme in their cases.

> "If there is no loan from the microcredit scheme, I would stay in the previous residence for longer time . . . " Interviewee 3

> "I have to move because of the removal order. If I did not receive the loan, I would be under huge financial stress to support the living expenses during the Chinese New Year". Interviewee 4

> "I don't have enough money for the relocation. If there is no such scheme, I probably would borrow money from my friends as the relocation was very urgent or I have to stay a while to save for it". Interviewee 7

Their responses and actions suggest that the cost of relocation is a financial burden in their relocation decisions, and the microcredit-like relocation assistance scheme helps them overcome financial difficulties for emergency and involuntary relocations in particular. Without financial support, they would usually choose to stay longer (resulting in a more serious spatial mismatch). Specifically, when it involves an emergency or involuntary relocation, the reason for relocation is about health and safety or the government's urgent removal orders, then a microcredit loan could make a big difference between going homeless or relocating to a habitable unit. In contrast, residents in the three controlled cases who do not require the microcredit to relocate are due to the fact that they obtained support from the Rainbow Fund or were allocated a public rental housing flat. These cases show that some low-income households could not relocate to a better home due to financial constraints, among other reasons. Sometimes even a small amount of relocation fees, such as the rental deposit, can deter them from deciding to relocate. However, when the financial constraints of the households are mitigated by means of microcredit, they will choose to relocate and become more residential mobile. This quasi-experiment study confirms the role of microcredit in the empowerment of residents of informal housing in residential mobility. This case study helps provide some initial evidence to address the concern of endogeneity and self-selection biases in explaining residential mobility by financial constraints raised by [11,12]. The results also extend Biggers et al.'s findings [5] that microcredits can assist low-income households in attaining residential mobility.

## 5. Conclusions

This study is a novel attempt to use an intervention approach by introducing a novel microcredit scheme for low-income SDU renters in Hong Kong to study whether their residential mobility is affected by financial constraints. The microcredit scheme is designated to assist low-income households in moving homes. It is the first microcredit scheme for the poor in Hong Kong. The samples include nine low-income households living in SDUs, which are subdivided from apartments to smaller-sized units. The SDUs are usually in poor conditions with irregularities but of higher unit rent per square meter than their counterparts in private formal housing markets. There have been many studies on the physical conditions of SDUs and their impacts on the well-being of the residents, yet there are very few studies on the residential mobility and financial constraints of SDU renters. This study interviewed the first batch of applicants of the microcredit-like relocation assistance scheme to investigate how it affects their relocation decisions. The results suggest that the availability of urgent financial support empowers them to move out of uninhabitable environments, especially in emergency or involuntary relocations. Most of the cases involved poor housing qualities or urgent relocation needs, such as water seepage, water pipe bursting, or government removal order. These are uncommon scenes in formal housing markets, but they are more frequently encountered in informal housing. Microcredit schemes for moving homes may not be crucial for the median to high-income households as they usually have savings and could easily borrow money from banks. However, this study shows that a microcredit scheme can be critical to the relocation decisions of low-income households and can even be a dividing line between homelessness and being sheltered. Since the microcredit scheme is a pilot scheme, the sample size of this study is small. It requires further larger-scale studies to empirically test the hypothesis. However, the responses and the actions of the respondents suggest that the residential immobility of low-income households caused by the financial constraints of the households can be mitigated by a microcredit scheme.

The results of this study offer important policy and practical implications. This quasi-experiment provides some evidence to support the provision of microcredit to residents of informal housing, which empower the vulnerable to have better bargaining power and improve residential mobility. It can help low-income households to avoid unnecessary risks and spatial lock-in. The pandemic raises the need for microcredit for residents of informal housing to an alarming proportion, and microcredit has been shown to be one of the future solutions to help low-income households to survive better in unaffordable cities.

It is recommended that governments and NGOs in unaffordable housing cities consider exploring the use of microcredit schemes to help households living in informal housing. It complements and is different from other charity schemes in supporting low-income households as the grantees are required to repay the funding organizations in microcredit schemes, which enables more sustainable support to the households without relying too much on new funding sources. It also helps make cities and human settlements more inclusive, safe, resilient, and sustainable (SDG 11) by empowering households to be able to choose where to live and mitigate spatial mismatches.

**Author Contributions:** Conceptualization, C.-Y.Y.; methodology, C.-Y.Y. and K.-M.L.; validation, C.-Y.Y. and K.-M.L.; formal analysis, C.-Y.Y. and K.-M.L.; investigation, C.-Y.Y. and K.-M.L.; resources, C.-Y.Y.; data curation, K.-M.L.; writing—original draft preparation, C.-Y.Y. and K.-M.L.; writing—review and editing, C.-Y.Y. and K.-M.L.; supervision, C.-Y.Y.; project administration, K.-M.L. All authors have read and agreed to the published version of the manuscript.

**Funding:** This research received no external funding.

**Institutional Review Board Statement:** Not applicable as the interviews were arranged by the NGO: HKSKH Lady MacLehose Centre Group & Community Work Unit.

**Informed Consent Statement:** Informed consent was obtained from all subjects involved in the study.

**Data Availability Statement:** Not available.

**Acknowledgments:** We would like to express our sincere appreciation for the assistance in data collection and organizing interviews by the social workers of the Hong Kong Sheng Kung Hui Lady MacLehose Centre.

**Conflicts of Interest:** The authors declare no conflict of interest.

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
