# Peer review of "The Roles of Microcredit in Informal Housing in the Future—A Case Study in Hong Kong"

_urbansci, doi:10.3390/urbansci6040091_

Round 1
Reviewer 1 Report
The paper contains new and significant information adequate to justify publication. However, there is a clear divergence between the literature review and the discussion of the analysis section. The discussion of the analysis section should have been integrated as a confirmation.
There is no need to reference self. Nevertheless, this is important, and please check the correct self-referencing style. The sample drawing can be used in the context of an example of an SDU. The note section should be part of the description of the figure or added to the footnote.
There is no need to separate the background context from the literature review—most of the information from part of the review document.
You need to include the regulatory standard for SDU and how the standard is different from what is available in the case study area.
The research should be clearly linked to Sustainable Development Goals (SDG) to provide further justification for the paper.
There is no clear statement for the hypothesis, yet this was discussed. If the paper is to test any hypothesis, state this clearly and include the test in the methodology. The methodology is unclear, and a clear justification of why the method has been adopted. The type of questions posed and the reason for the question are not included in the paper. The information used in the discussion of analysis, such as the reason for moving, effects of the microcredit scheme on borrowers’ rental decisions and the variables identified though implied in the literature review section, is not clearly represented in the analysis section, making it difficult to ascertain how conclusions are drawn. Whilst this paper affirms the novelty of the research, it is clear how this is represented in the paper. For instance, the recommendation for low-income households to explore the use of microcredit schemes for SDU should include the requirements and accessibility factor for using this type of scheme and why it is better than other types of schemes available in Hong Kong.
Finally, the paper should be proofread before revision submission.
Reviewer 2 Report
This case report is meaningful. Please see the attachment for my detailed comments and suggestions.

Round 2
Reviewer 1 Report
The authors have significantly improved on work however require some minor additions and clarifications.
There are two figures represented in the study, but one was mentioned or referred to in the study. If the two figures represent the same information, I suggest you put them side by side and minimise the size, also stated clearly in the footnote and description of the figure section.
The discussion section has impliedly referred to in the literature review section. However, no citations were included to reflect integration.
Reviewer 2 Report
Nice work. I carefully reviewed the author's responses and all 13 questions were answered in detail and appropriately. I think these changes will make a strong impact on what the potential readers take from this paper.
I recommend this paper be accepted in its present form.
Author Response
Thank you very much for accepting our manuscript.
It is also proofread to correct the minor grammatical errors.